## Lagrangian characterization of nitrate supply and episodes of extreme phytoplankton blooms in the Great Australian Bight

Paulina Cetina-Heredia<sup>1,2</sup>, Erik van Sebille<sup>1,2</sup>, Richard Matear<sup>3</sup>, and Moninya Roughan<sup>4</sup>

<sup>1</sup>Climate Change Research Centre, ARC Centre of Excellence for Climate System Science, UNSW Australia, Sydney Australia

<sup>2</sup>Grantham Institute & Department of Physics, Imperial College London, London, United Kingdom

<sup>3</sup>CSIRO Marine and Atmospheric Research, CSIRO Marine Laboratories, Hobart, Australia

<sup>4</sup>Regional and Coastal Oceanography Laboratory, School of Mathematics and Statistics, UNSW Australia, Sydney, Australia

Correspondence to: Paulina Cetina-Heredia (p.cetinaheredia@unsw.edu.au)

Abstract. Phytoplankton growth is the foundation for energy transfer into higher trophic levels, influences climate by the uptake of atmospheric  $CO_2$ , and plays an important role in nutrient cycling. Here we use a novel Lagrangian approach to characterize the nitrate supply to the Great Australian Bight, identify episodes of extreme phytoplankton blooms and ascertain the origin of the nitrate sources that fuel them. We find that 55% of nitrate used by phytoplankton enters the GAB in the

5 upper 100m and that 88% originates locally from a region between the GAB and the Sub Antarctic Front, rather than from more remote oceans; thus, most of the nitrate is recycled locally. Our results show extreme phytoplankton blooms have an annual periodicity, peaking in the Austral autumn when the mixed layer deepens. This suggests that stratification erosion is key supplying nutrients into the euphotic zone and triggering these episodes.

### 1 Introduction

- Primary productivity, or the rate at which phytoplankton converts energy to organic matter, regulates two of the largest anthropogenic impacts on marine ecosystems: fishing and climate change. Specifically, it sustains higher trophic levels influencing the health of ecosystems and ultimately fisheries (e.g. Pauly and Christensen, 1995; Chassot et al., 2010); and plays an important role in atmospheric  $CO_2$  uptake; the phytoplankton and utilized carbon eventually sink into the ocean interior or are grazed by zooplankton (i.e. biological pump, e.g. Falkowski et al., 1998; Behrenfeld et al., 2006). Characterizing phytoplankton growth
- is therefore essential due to globally declining trends in fisheries catchment recorded since the twentieth-century (Pauly et al., 2005) and ongoing increases in atmospheric  $CO_2$  concentrations (e.g. Solomon et al., 2009).

Coastal regions account for a significant part of the oceans productivity (Pomeroy, 1974; Gattuso et al., 1998). The Great Australian Bight (GAB) is a highly productive semi-enclosed sea along the coastline of southern Australia that sustains the largest finfish fishery in Australia (McClatchie et al., 2006) and has a unique biodiversity with 85% of the species being

endemic (Petrusevics et al., 2009). Episodes of phytoplankton growth and productivity within the eastern side of the GAB have been linked to wind driven upwelling episodes in summer (e.g. Kämpf et al., 2004). However, a thorough characterization of

spatial and temporal variability, up to inter-annual, of phytoplankton blooms, or episodes of extreme phytoplankton growth in the GAB is lacking.

Phytoplankton growth is regulated by multiple factors including light, temperature, and nutrient availability (Boyd et al., 2010). Nitrate and iron are the most common nutrients limiting phytoplankton growth globally (Moore et al., 2013). Iron

- supply from sediments is greatest over the continental shelves (Wadley et al., 2014); thus, its concentration is typically higher closer to the coast (Graham, Robert M et al., 2015) and likely to be available in the GAB due to large deposits of iron ore in the coast (http://www.ga.gov.au/corporate\_data/70156/70156.pdf). In contrast, high nitrate concentrations occur deep in the open ocean, for instance, in Southern Ocean waters; however, it remains unknown if the supply of nitrate into the GAB occurs at depths where light penetrates. Moreover, (Irwin et al., 2015) have shown that phytoplankton cannot adapt to changes in nitrate
- availability. Therefore, this study examines the supply of nitrate into the GAB. Using Redfield ratios, our findings can be used to infer other nutrient sources such as phosphate. We focus on episodic phytoplankton blooms, a fundamental feature of phytoplankton population dynamics that has important implications on organisms that feed on phytoplankton, biogeochemical cycles, and water quality (Cloern, 1991, 1996).
- The GAB receives water from adjacent oceans (i.e. Pacific, Indian, Southern) with different nitrate loads and it is unclear where the nitrate that is used for primary productivity in the GAB comes from, the mechanism that makes it available for phytoplankton uptake, and how that changes over time. For instance, the influence of seasonal and longer-term fluctuations of currents that feed water into the GAB on phytoplankton growth within the GAB is unknown. Here, we use a Lagrangian approach for the first time to identify the origin of the nitrate utilized by phytoplankton and identify episodes of extreme phytoplankton blooms. By associating the extreme phytoplankton blooms to sources supplying nitrate, and characterizing their
- spatial and temporal variability over a 12-year period, we can disentangle which mechanisms may induce high productivity inside the GAB.

#### 2 Method

#### 2.1 Model

- Outputs of a biogeochemical ocean model are used to track water parcels going into the GAB and examine changes of nitrate and phytoplankton concentrations along their trajectories. The Whole Ocean Model of Biogeochemistry And Trophic-dynamics (WOMBAT; Matear et al., 2013) is a Nutrient, Phytoplankton, Zooplankton and Detritus (NPZD) model that has been incorporated into the hydrodynamic, eddy-resolving, Ocean Forecast Australian Model (OFAM; Oke et al., 2008). OFAM has 47 vertical layers and a spatial resolution of 1/10° that yields an adequate modeling of currents around Australia (Sun et al., 2012). Moreover, Matear et al. (2013) showed that features influencing phytoplankton growth through the supply of nitrate, such as the
- mesoscale structure of currents and eddies in the Tasman Sea (Pacific Ocean), are well captured in OFAM-WOMBAT outputs. In addition, modeled spatial patterns and seasonal evolution of phytoplankton in the Tasman Sea and Southern Ocean agree with satellite observations (Matear et al., 2013). Therefore, OFAM-WOMBAT outputs are ideal to show how the Lagrangian approach can be used to determine nitrate supply and phytoplankton dynamics. Comparisons of horizontal and vertical distri-

butions of modeled nitrate concentrations with observations compiled into the CSIRO Atlas of Regional Seas (CARS, released in 2009; Ridgway and Dunn, 2003) show that the observed spatial distribution of low nitrate concentrations close to the coast and high nitrate concentrations in the Southern Ocean is replicated in the model (Figure 1). However, the model depicts more spatial variability than that of the CARS observations. This may in part be due to the limited observational coverage and sub-stantially coarse resolution of CARS, which makes CARS overly smooth. The model captures well the order of magnitude

5

stantially coarse resolution of CARS, which makes CARS overly smooth. The model captures well the order of magnitude, variability, and change over depth of annual mean nitrate concentrations at specific locations in the Pacific, Southern, Indian oceans, and the region between the GAB and the SAFn (Figure 1).

#### 2.2 Lagrangian approach and simulations

Using Lagrangian or combined Eulerian-Lagrangian approaches to investigate ocean biogeochemistry has a number of advantages (Chenillat et al.). For instance, an increase in phytoplankton concentrations along a water parcel trajectory (Lagrangian) corresponds to phytoplankton growth, whereas an increase in phytoplankton concentration at a fixed location can also be due to advection of biomass into the fixed observational frame (Jönsson et al., 2011). Thus, to diagnose phytoplankton growth and identify the sources of utilized nitrate, we examine temporal changes of nitrate and phytoplankton concentration along water parcel trajectories.

- Backward and forward water parcel trajectories are used to respectively reveal: 1) where the water and nitrate that reach the GAB come from and at which depth it enters, and 2) where inside the GAB and when is nitrate utilized by phytoplankton for rapid growth. Water parcel trajectories are simulated with a Lagrangian tracking algorithm, the Connectivity Modeling System (CMS; Paris et al., 2013) using three-dimensional daily velocities modeled with OFAM-WOMBAT, and seeding 'particles' daily from June1993-June2005, along the 2000m isobath delimiting the GAB, (Figure 1), throughout the water column, in grid
- cells spaced every  $1/10^{\circ}$  degree horizontally and 10m in the top 200m. Particles are only released at times and locations when the instantaneous flow advects water parcels into the GAB; thus, the positions and spacing between seeded particles varies daily. This results in a total of approximately  $2.5 \times 10^6$  particle trajectories at an average of ~ 600 particles released per day. The position (longitude, latitude and depth) of the water parcels is recorded daily. OFAM-WOMBAT nitrate and phytoplankton concentrations are interpolated onto the water parcel trajectories and used to determine the nitrate memory time, and episodes 25 of simultaneous extreme nitrate decrease and phytoplankton increase (see sections below).

#### 2.3 Nitrate memory time & sources

To deconstruct the contribution of nitrate from different oceans into the GAB, we quantify the time over which the nitrate concentration of water parcels that reach the GAB prevails along their backward trajectories (i.e. de-correlation time). This can be considered a memory time of nitrate concentration. If the nitrate concentration of a water parcel becomes significantly

different over time along its trajectory it means that the resource has been used or replenished; thus, short memory times imply local resource recycling. Therefore, the nitrate source of water masses entering the GAB is the location of the water parcel just before its nitrate concentration loses its correlation to the nitrate concentration of the water mass at the time it enters the GAB. The nitrate memory time or de-correlation time is obtained following Emery and Thomson (2001), by computing the

correlation between the anomaly of nitrate concentration when the water parcels enter the GAB, and the anomaly of nitrate concentration at every time step before reaching the GAB along the water parcel trajectories (Figure 2). The last time when the correlation is larger than zero corresponds to the de-correlation time scale or nitrate memory time. Our results show that nitrate memory is lost after  $82 \pm 0.21$  days for the entire set of  $2.5 \times 10^6$  particles (Figure 2).

- 5 The location of each water parcel 81 days before it crosses the 2,000m isobath entering the GAB is used to determine whether the nitrate source is either the Pacific, Southern, Indian Ocean, or a region between the GAB and the Subantarctic Australian Front as defined by Sallée et al. (2008) using sea surface height and temperature gradients (GAB-SAFn, Figure 2). Similarly, the depth (above or below 100m) at which the water parcel enters the GAB is recorded. Vertical profiles of the phytoplankton concentration gradient inside the GAB show a peak around 100m where phytoplankton decreases rapidly with
- 10 depth due to light limitation on phytoplankton growth (Figure S1); by choosing 100m as a threshold we distinguish between nitrate entering the GAB within or below depths were it is readily used.

# 2.4 Episodes of extreme phytoplankton blooms from simultaneous extreme nitrate decrease and phytoplankton increase

In WOMBAT, the change of phytoplankton concentration depends upon the efficiency of phytoplankton to uptake nutrients, their availability and the grazing by zooplankton; as a consequence, when phytoplankton increases nitrate decreases. We compute the difference in nitrate and phytoplankton concentration every time step along forward trajectories of water parcels after they enter the GAB and identify instances when a decrease in nitrate and increase in phytoplankton concentrations are great, hereafter referred to as extreme phytoplankton blooms. Specifically, these episodes are defined as times when the change in nitrate is negative, the change in phytoplankton is positive, and both are extreme (i.e. top 5%) of their respective

- distributions. Not in all instances of extreme nitrate decrease is the phytoplankton increase also extreme and vice-versa; The extreme phytoplankton blooms identified in this study account for 14% and 20% of instances when decrease and increase in nitrate and phytoplankton are in the top 5% of their distributions respectively. The extreme phytoplankton blooms represent 0.35% of all data (i.e. total of daily changes in nitrate and phytoplankton across all particle trajectories). The locations and time of these extreme phytoplankton blooms are recorded, as well as the source of the water parcel and the depth at which it
- 25 entered the GAB. The changes in nitrate and phytoplankton concentrations of every episode were used to construct: 1) maps showing the spatial distribution of extreme phytoplankton blooms and their intensity (i.e. means of daily change in nitrate, and phytoplankton across episodes (Figure 3), and 2) time series of change in phytoplankton across extreme phytoplankton blooms for further analysis (Figure S2).

#### 2.5 Temporal variability

30 A wavelet analysis is conducted on the time series of the change in phytoplankton concentrations across extreme phytoplankton blooms following Torrence and Compo (1998) and Liu et al. (2007). We focus on time series of change in phytoplankton rather than nitrate because we are ultimately interested in extremes in primary productivity; nevertheless, as previously mentioned, the examined extreme phytoplankton blooms are only those when extreme changes occur in both phytoplankton and nitrate

concentrations. Time series of changes in phytoplankton concentration of extreme phytoplankton blooms can be decomposed by location within the GAB where they occur (longitudes), by nitrate source, and by depth at which the water parcel entered the GAB (Figure S2). Wavelet analysis on each constituent of the time series is conducted to reveal periodicities linked to specific regions inside the GAB where the extreme phytoplankton blooms occurred, nitrate sources, and depths at which the water parcels enter the GAB (Figure 4, Figure S4).

5

#### 3 Results

#### 3.1 Nitrate memory time & sources

A map showing the minimum time it would take a particle to reach the GAB given its location is constructed from all the water parcel trajectories (Figure 2b). This maps shows that in the nitrate memory time (81 days, see section 2.3) most of the
particles entering the GAB are likely to be a few 100km off the shelf, predominantly in a region between the GAB and the SAFn. However, the ~ 81 day contour does extend into the Pacific, Indian, and Southern oceans and therefore, although much less probable, some water from these oceans can reach the GAB within the nitrate memory time (Figure 2).

Similar volumes of water that reach the GAB and remain inside it for at least 81 days enter the semi-enclosed sea above 100m and between 100–2000m; however, most of the nitrate input occurs below 100m due to higher nitrate concentrations at

15 depth (Table 1). Despite larger nitrate input occurring below 100m,  $\sim 55\%$  of the extreme phytoplankton blooms are linked to water entering the GAB in the upper 100m; moreover, monthly means of daily phytoplankton increase linked to water entering the GAB in the top 100m are of similar or larger magnitude ( $0.5 - 4.6 \mu mol m^{-3} d^{-1}$ ) than those linked to water entering the GAB between 100–2000m ( $0.05 - 7 \mu mol m^{-3} d^{-1}$ , Figure 4d, Figure S3h-i).

The volume of water that reaches the GAB from the Indian Ocean (~5%) is three times larger than that from the Southern 20 Ocean (~1.5%); however, nitrate concentrations are higher in the latter, making the contribution of the Southern Ocean to total nitrate input into the GAB only half that of the Indian Ocean (Table 1). Similarly, 10% of the extreme phytoplankton blooms are linked to water coming from the Indian Ocean, while only 2% are linked to the Southern Ocean; nevertheless, the monthly means of daily increase in phytoplankton are the same order of magnitude (~ 1 µmol m<sup>-3</sup> d<sup>-1</sup> Figure 4c, Figure S3e-f) irrespective of the nitrate source. Negligible percentages (<1%) of water and nitrate that enter the GAB as well as extreme</p>

25 phytoplankton blooms (~ 0.1%) can be tracked back or linked to the Pacific Ocean, respectively (Table 1). These results are in agreement with higher connectivity between western and southern Australia – induced by prevailing eastward surface flow – than between eastern and southern Australia (Coleman et al., 2013).

## 3.2 Episodes of extreme phytoplankton blooms from simultaneous extreme nitrate decrease and phytoplankton increase

Biogeosciences Discussions

#### 3.2.1 Spatial distribution & magnitude

Maps showing the horizontal spatial distribution of extreme phytoplankton blooms show the largest daily changes in nitrate and 5 phytoplankton over the shelf break (Figure 3) suggesting that topographically induced uplift is a key mechanism allowing deep nitrate rich water to reach the euphotic zone and trigger productivity. In addition, vertical positions of forward trajectories of water that enters the GAB show that 94% of the extreme phytoplankton blooms involve upward displacement of water masses, and 56% of the extreme phytoplankton blooms are associated with water uplift over at least 5 m d<sup>-1</sup> and up to tens of meters per day.

- 10 Means (across extreme phytoplankton blooms occurring at each location in Figure 3 map) of daily changes in phytoplankton concentration associated with the extreme phytoplankton blooms ( $\sim 0.04 - 1 \,\mu mol \,m^{-3} \,d^{-1}$ ) are often an order of magnitude smaller than those in nitrate ( $\sim 0.5 - 10 \,\mu mol \, m^{-3} \, d^{-1}$ ) due to a constraint in the efficiency of phytoplankton to uptake nitrate and transform it into biomass which reflects phytoplankton losses from grazing and mortality. Substantial changes (i.e. means larger than 1 and  $-0.1 \,\mu\text{mol}\,\text{m}^{-3}\,\text{d}^{-1}$  for nitrate and phytoplankton respectively) are aligned with the 500m isobath, and peak south of Eyre Peninsula (34°S, 135°E) in the eastern side of the GAB. 15

#### **Temporal variability** 3.2.2

From May–Jul, extreme phytoplankton blooms occur mostly in the centre of the GAB (125–135°E longitude), along trajectories of water that entered the GAB throughout the water column but more often below 100m, and that carried nitrate from a region between the GAB and the SAFn (Figure 4). Conversely, from Sep-Apr, extreme phytoplankton blooms occur almost exclusively in the west (107-124°E longitude) and east (136-150°E longitude) sides of the GAB, along trajectories of water that entered the GAB in the upper 100m, some of which carried nitrate from farther, mostly from the Indian and Southern

Oceans (Figure 4 & Figure S3). Means of daily increase in phytoplankton concentration associated with extreme phytoplankton blooms mostly peak in May and October (Figure 4).

A wavelet analysis of the time series of the extreme phytoplankton blooms reveals  $\sim 15$  day, semiannual, and annual peaks (Figure 4). The magnitude (i.e. significance and dominance) of the semiannual and annual periodicities associated with extreme 25 phytoplankton blooms varies with the longitude where the extreme phytoplankton blooms occur as well as the nitrate source, and depth at which the water parcels enter the GAB (Figure 4e-f). The occurrence of extreme phytoplankton blooms is largely confined to the Austral autumn from Apr-Jun, and spring from Sep-Nov (Figure 4, Figure S3).

#### 4 Conclusions

20

This study uses a lagrangian approach to quantify for the first time the origin and memory time of nitrate that enters the GAB. 30 We found a nitrate memory time of 81 days for water parcels reaching the GAB. This implies that over this time scale either

biological processes such as nitrate uptake and remineralization, or physical processes other than advection are likely to induce substantial changes in nitrate concentrations. Due to the short timescale, most (93%) of the nitrate advected into the GAB comes from a region between the GAB and the SAFn and little of the nitrate signal (7%) can be attributed to water coming from further afield. This implies that most of the nitrate is recycled regionally. The adopted lagrangian approach also allows identifying extreme phytoplankton blooms where there is simultaneous extreme decrease in nitrate and extreme increase in

phytoplankton within the GAB.

The mean daily increase in phytoplankton across all extreme phytoplankton blooms (0.09  $\mu$ mol m<sup>-3</sup> d<sup>-1</sup>) is of similar order of magnitude as phytoplankton growth rates in productive environments. For instance, Falkowski et al. (1991) measured growth rates of ~ 0.06  $\mu$ mol m<sup>-3</sup> d<sup>-1</sup> inside a cold core eddy where nutrient rich water is constantly supplied into the euphotic layer

- through upwelling. The extreme phytoplankton blooms account for 6% net primary productivity but represent less than 1% of the particle trajectories showing a daily nitrate decrease. In total, the particles trajectories estimate a net daily nitrate of uptake in the GAB of (0.069  $\mu$ mol N m<sup>-3</sup> d<sup>-1</sup>) of which about 6% requires a source of nitrate from outside the GAB. Thus, the GAB is a productive region, which is primarily driven by recycled nitrate.
- The locations of extreme phytoplankton blooms align remarkably with the shelf break, suggesting that topographical uplift
  plays a crucial role in supplying nitrate to the euphotic zone and provoking the blooms. In fact, the interaction with the bottom of the subsurface westward flowing Flinders Current is upwelling favourable (Middleton and Bye, 2007), and therefore likely to facilitate the uplift of nutrients particularly along the 600m isobath, where the Flinders Current is centered, and inshore. Similarly, Herzfeld and Tomczak (1997) found that flow strengthening and consequent bottom stress intensification west of Eyre Peninsula (west of 135°E) induces bottom boundary layer upwelling, which exceeds upwelling driven by the winds
  (Middlaton and Pletory 2003)
- (Middleton and Platov, 2003).

Although most of the nitrate input into the GAB occurs below 100m (Table 1), roughly half of the extreme phytoplankton blooms are associated with water that enters the GAB above 100m. These results suggest that despite uplift, not all the water parcels entering at depth reach the euphotic zone where phytoplankton thrives, and that low stratification is required to supply nitrate to depths where light is sufficient for phytoplankton growth. Indeed, the extreme phytoplankton blooms linked to deep

- nitrate rich water occur mostly in autumn when enhanced winds and surface heat loss reduce stratification (Figure 4d), and a deepening of the mixed layer to ~ 60m has been observed (Condie and Dunn, 2006); particularly, in the centre of the GAB. Conversely, episodes associated with shallow water occur year-round and peak in spring (Figure 4d) when stratification starts to re-establish (Condie and Dunn, 2006). Our findings indicate that low stratification and topographically induced uplift are therefore key for producing the extreme phytoplankton blooms.
- The extreme phytoplankton blooms are evidence of canonical spring-autumn blooms described by Longhurst (1995) and Cloern (1996). This pattern usually emerges as a result of stratification breakdown in autumn, which makes nutrients accumulated below the summer pycnocline available for uptake, and seasonal increase in solar radiation during spring (e.g. Cushing, 1959). A summertime wind driven upwelling system (Great South Australian Coastal Upwelling System) between Eyre Peninsula and Bonney coast (~ from 135–142°E longitude) has been identified and linked to primary productivity (Kämpf et al.,
- 2004). Although we found extreme phytoplankton blooms on the east side of the GAB during summer months (Figure 4, Fig-

5

ure S3a); they are considerably fewer and less intense than those occurring in autumn and spring. Thus in the model, vertical mixing, topography, and light exposure seem to be more relevant than winds in producing extreme phytoplankton blooms. In agreement with these results Kämpf (2015) found that strong upwelling-favorable winds do not always induce phytoplankton blooms; in addition, they found that during summer, extreme increments in chlorophyll a concentrations (i.e. above 2.5  $mg m^{-3}$ ) are infrequent.

Most of the instances when nitrate sources are attributed to a specific ocean correspond to episodes that take place in late winter and spring (Aug-Nov). This is likely due to a winter intensification of the Leeuwin Current which flows polewards along the west coast of Australia and turns east around Cape Leeuwin bringing water from the Indian Ocean into the GAB within a timeframe over which nitrate is, according to our results, correlated (i.e. 82 days). Indeed, Herzfeld and Tomczak

(1997) identified the intrusion of the Leeuwin current in the GAB in winter. Similarly, seasonal wind speeds between Australia 10 and Antarctica are stronger in spring (e.g. Luis and Pandey, 2004) potentially increasing the flow from the Southern Ocean into the GAB through Ekman transport, and leading to episodes in spring that are linked to the Southern Ocean.

Our results reveal inter-annual variability in the intensity of extreme phytoplankton blooms are associated with non-local sources. Firstly, the annual periodicity of episodes linked to the Southern Ocean is significant from 2000 onward (Figure S4).

- 15 This may be the result of a positive trend of the Southern Annual Mode (SAM) in the last decades (e.g. Marshall, 2003; Sallée et al., 2010) which has the potential to facilitate the arrival of typically nutrient rich Subantarctic Mode Water (SAMWs) from the Southern Ocean into the GAB through a southward shift and intensification of the southern hemisphere westerlies (e.g. Thompson and Wallace, 2000). Ayers and Strutton (2013) have shown a correlation between the nutrient load in SAMWs and the SAM within the Pacific sector of the Southern Ocean through enhanced upwelling due to wind field changes induced by
- a positive SAM. Focusing on extreme phytoplankton blooms our study suggests that such a mechanism is also in place in the 20 Australian sector. Secondly, a significant annual periodicity between 1998-2001 of extreme phytoplankton blooms are linked to the Indian Ocean (Figure S4) may be a consequence of the Leeuwin current intensification during La Niña years (e.g. Feng, Ming, 2003). Finally, there are no significant inter-annual fluctuations in extreme phytoplankton blooms linked to water coming from the Pacific, this is likely a consequence of the negligible contribution of Pacific waters into the GAB.
- Author contributions. E. van Sebille and P Cetina-Heredia devised the aims of the study and methodology; P. Cetina-Heredia conducted 25 data analysis; R. Matear produced and provided the outputs of the biogeochemical model. All authors contributed to the interpretation of the results and the writing of the manuscript.

Acknowledgements. OFAM-WOMBAT is produced by CSIRO, BoM, and the Royal Australian Navy and provided in http://www.marine.csiro.au/ofam1/. Torrence and Compo [1998] wavelet software is provided in URL:http://atoc.colorado.edu/research/wavelets, the bias correction by Liu et al.

(2007) is available at ocg6.marine.usf.edu/\_liu/wavelet.html. This work was funded by the ARC via grant DE130101336. We thank colleagues from the UNSW CCRC, and the Coastal and Regional Oceanography Lab for their feedback.

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

**Table 1.** Contribution (%) from each source to: 1) water volume going into the GAB and 2)  $NO_3$  input from June 1993 to June 2005, categorized by the depth at which it enters the GAB.

|       | Pacific           |                    | Indian            |                    | Southern          |                    | GAB-SAFn          |          | All sources       |          |
|-------|-------------------|--------------------|-------------------|--------------------|-------------------|--------------------|-------------------|----------|-------------------|----------|
|       | water             | [NO <sub>3</sub> ] | water             | [NO <sub>3</sub> ] | water             | [NO <sub>3</sub> ] | water             | $[NO_3]$ | water             | $[NO_3]$ |
|       | volume            | (µmol N)           | volume            | (µmol N)           | volume            | (µmol N)           | volume            | (µmol N) | volume            | (µmol N) |
|       | (m <sup>3</sup> ) |                    | (m <sup>3</sup> ) |                    | (m <sup>3</sup> ) |                    | (m <sup>3</sup> ) |          | (m <sup>3</sup> ) |          |
| <100m | 0.14              | 0.07               | 4.64              | 0.86               | 1.57              | 0.51               | 35.31             | 10.73    | 41.66             | 12.17    |
| >100m | 0.24              | 0.38               | 0.24              | 0.13               | -                 | -                  | 57.86             | 87.31    | 58.34             | 87.83    |
| All   | 0.38              | 0.46               | 4.88              | 0.99               | 1.57              | 0.51               | 93.17             | 98.04    |                   |          |