# Peer review of "Lagrangian characterization of nitrate supply and episodes of extreme phytoplankton blooms in the Great Australian Bight"

_Biogeosciences, 2016_

## Referee Comment (RC1) · B.F. Jonsson (Referee) · 18 Apr 2016

Tha authors use output from a 1/10° GCM with biogeochemistry to asses sources of nitrate into the Great Australian Bight. They connect the input of nitrate to episodic blooms in the model. The analysis is conducted by applying Lagrangian particle tracking.

The paper has great potential and show very interesting results. I have, however, some minor questions about the methodology and there are further analysis that might strengthen the authors' statements.

The study is entirely based on model data. It would be good to see a validation of the

results from observations for example from remote sensing. Such data would show conclusively if/where/when the reported modeled production events occur in real life. There must be plenty of cruises with north-south transects in the area? If so, maybe you could compare the decrease in NO3 along trajectories with the change between stations from such cruises.

It seems to me from figure 1 that the model significantly overestimates the drawdown of NO3 in the central part of the study domain. This difference needs to be explored since it can dramatically change the relative importance of different NO3 sources. Also, please use either hotter colors for high values or a truly sequential colormap in figure 1a. I thought for the longest times that green areas had lower levels of NO3 than yellow areas.

Page 3 line 27 — page 4 line 11. I didn't follow how you calculated the decorrelation time of NO3. My interpretation is probably wrong since it doesn't make any sense, but It seems like you don't take absolute concentrations of source NO3 into account? I'd recommend that you make this section more explicit, and that you write out how you apply the method from Emery and Thomson better.

Page 4 lines 14-28. It would be very useful to compare these values with growth rates from the model, which should be explicitly calculated by the NPZD module.

Page 5 lines 8-27. How do you take vertical diffusion of nutrients into account? I would assume that a significant amount of tracer below 100 meters could be transported in ways not picked up by the particle tracking?

How much of the NO3 in GAB/SAFn as SAF originates from the southern or indian ocean? It's possible that some of the available nutrients in nearby areas in fact has been transported there from afar.

---

## Referee Comment (RC2) · Anonymous Referee #2 · 5 May 2016

General Comments

The authors study "extreme phytoplankton blooms" by analyzing model output to infer the source of nitrogen (geographic region, depth), timing, and location of these events. The motivation for the study is not clear: either the focus on the most rapid increases in chlorophyll or the consequences of knowing the sources of N consumed in these blooms. Their conclusions are that much of the N for these extreme blooms originates in the upper 100 m of the ocean between the Great Australian Bight and the Sub-Antarctic front. While this is an interesting analysis, the significance is not clearly communicated.

Specific Comments

The paper is generally easy to understand, but key ideas are frequently undeveloped (why focus on extreme blooms). This particularly problematic in the discussion which I found to consist of many briefly mentioned and not clearly connected ideas.

Would the conclusions of the paper differ if the authors focussed on less extreme phytoplankton blooms? How much of total productivity is in these extreme blooms? (I see this is at page 7, line 10; an earlier motivational mention would be helpful.) Blooms take days to weeks to develop, why do you focus on such short periods (large daily changes)? Perhaps large growth rates would be worth a closer look?

I'm going to echo two comments by the other referee: (1) the study is based entirely on model calculations; some connection to remote sensing data or other observations would be most welcome, (2) the description of decorrelation / memory time should be expanded.

Technical Corrections

Abstract - "novel Lagrangian approach" - what is the value of stressing the novelty? You don't clearly explain why its novel.

"enters the GAB in the upper 100m" – sounds like this is a contrast with "stratification erosion is key supplying nutrients". Stratification erosion does not actually supply nutrients. Are you referring to entrainment of nutrients from below the nutricline? Which is it? Lateral advection or entrainment?

p1, l10. I think "regulates" is an overstatement.

p1, l15. This is an unusual use of the word "catchment".

p1, l17. "Coastal regions account for a significant.." this is vague. Can you be more precise?

p2, l12. "important implications": this is vague. What are the implications of "extreme blooms" (here called episodic) that justify the focus in this study?

p2, l18. "first time": the focus on priority rather than your message is not helpful or needed.

p2, l20. I didn't find a disentangling of the mechanisms later in the paper.

p3, l10. Year missing from citation.

p3, l25. There is some good detail in this paragraph, but I found that I still have considerable uncertainty about what you did. For example, how is memory time defined and calculated?

p3, l28. I didn't understand "prevails", and thus the whole sentence escaped interpretation.

p3, l29. This is another vague explanation of "memory time". A rewriting of this paragraph would be helpful.

p4, l9. Do you really mean that the concentration gradient has a peak (trough?) here? Why does light create a sudden increase (decrease?) in the concentration gradient?

p4, l11. "were" should be "where"

p4, l14. What do you mean by "efficiency" here?

p5, l25. I don't think you need "respectively" here.

p6, l10-15. I found this difficult to follow. if 90% of the N is disappearing, does that imply rapid grazing tightly coupled to very short term increases (daily) in phytoplankton biomass? Are you sure your grazing model is good enough to draw these conclusions and make them interesting?

p8, l2. You compare to the effects of winds here, but I didn't see where you analyzed winds in your methods + results.

p8, l13-16. I found the focus on interannual features, which were not discussed elsewhere in the paper, to be jarring and confusing at this point. Might be very interesting /

important, but should appear earlier, with some evidence or mention of analysis.

---

## Author Comment (AC1) · 7 Jun 2016

We thank the reviewers for their comments. This is our response letter to reviewers (1&2). Find the reviewer comments in bold followed by our response.

*Reviewer 1*

**The study is entirely based on model data. It would be good to see a validation of the results from observations for example remote sensing. Such data would show conclusively if/where/when the reported model production events occur in real life. There must be plenty of cruises with north-south transects in the area? If so, maybe you could compare the decrease in NO3 along trajectories with the change in stations from such cruises**.

OFAM-WOMBAT, the model used in this study, is a well-established model configuration that has been previously validated against observations; particularly, Matear *et al.* 2013 compares chlorophyll outputs against satellite observations revealing that modeled spatial patterns and seasonal evolution of phytoplankton in the Tasman Sea and Southern Ocean agree with observations (Page 2 Lines 29-32).
We furthermore compare modeled and observed annual vertical profiles and depth averaged $NO_3$ fields (Page 2 Lines 33 Page3 lines 1-7) showing that the model depicts $NO_3$ distributions correctly.

Nevertheless, we agree with the reviewer that MODIS ocean color is a useful product to further evaluate the model simulation and the study results by comparing not only the spatial and temporal chlorophyll distributions as Matear *et al.* 2013 do, but those of phytoplankton blooms characterized in this study. One objective of the study is to characterize the temporal variability of extreme phytoplankton blooms; we found the model simulation has a bimodal temporal distribution with local maxima occurring in the austral spring and autumn (Figure 4a in the manuscript).
A similar pattern of chlorophyll extremes is also evident in the MODIS remotely sensed daily chlorophyll concentrations (Figure 1, response letter). For the model-data comparison, we computed the daily changes in chlorophyll concentration inside the GAB from 2003-2005 (period of time when satellite and modeled data coincide) using satellite data downloaded from: http://thredds.aodn.org.au/thredds/catalog/IMOS/SRS/OC/gridded/aqua/1d/catalog.html. From the daily changes we selected changes larger than the 95 percentile to compute extreme phytoplankton blooms in the GAB. The extremes were then averaged into monthly bins to provide a measure of the magnitude of the extreme events with season as: $\overline{\frac{D[phy]}{Dt}}_m = \frac{\sum_{i=1}^{n}\left[D[phy]/Dt\right]_i}{n}$, where D[phy]/Dt are changes in phytoplankton concentrations over one day (*Dt*), *i* is the day number, and *n* is the number of days in each month *m*. The resulting analysis shows the remotely sensed observations have a bimodal temporal distribution (Figure 1, response letter) consistent with the model simulation. The observations support the model analysis that the occurrence of phytoplankton blooms increase as one approaches the austral spring (Aug) and autumn (Mar)

reaching a maximum in May as we found from modeled data (Figure 1, response letter).

[Figure]

Figure 1. Monthly means of daily increase in phytoplankton associated with extreme phytoplankton blooms a) in the observations occurring between 2003-2005 and b) in the simulation occurring between June 1993 and June 2005.

In addition, we compared the spatial distribution of phytoplankton blooms from modeled phytoplankton and remotely sensed chlorophyll. In the observations it is not possible to select extreme phytoplankton blooms that are associated with extreme nitrate decrease because we lack nitrate observations. Therefore, for the comparison we constructed maps of extreme phytoplankton blooms using all extreme phytoplankton blooms irrespective of whether such blooms are associated with extreme nitrate loss or not (Figure 2, response letter). Both, the distribution and intensities of extreme phytoplankton blooms show similar patterns, where the large extreme blooms occur between the coastline and the 70m isobath with patches south of 36°S within the GAB-SAFn region. Discrepancies between satellite and modeled phytoplankton bloom intensities are to be expected because: 1) There are gaps in the observations due to cloud cover which could bias the remotely sensed results, 2) most importantly near the coast the extreme blooms may not reflect phytoplankton blooms because the satellite cannot separate phytoplankton from other optical contaminants like suspended sediment, and 3) The model is not well suited for shallow in shore environments where water is less than 50m. Further, in the shallow environment the simulated extreme phytoplankton blooms are never associated with extreme nitrate decreases (Figure 3, response letter). This suggests that the interaction of the water column with bottom sediments may be a constant source of nitrate, avoiding extreme decrease in nitrate despite high productivity. This would be included in the discussion.

[Figure]

Figure 2. Surface spatial distribution of extreme phytoplankton blooms from a) satellite observations from 2003-2005 and b) modeled outputs from June 1993 and June 2005.

[Figure]

Figure 3. Simulated horizontal distribution of episodes of extreme nitrate decrease occurring between June 1993 and June 2005. The map shows the mean of the daily decrease in nitrate concentration (absolute values).

In conclusion we have analyzed remote sensed daily chlorophyll data (provided daily) and found a similar spatial pattern in the extreme phytoplankton blooms to our simulation (i.e. high close to the coastline and in patches south of ~36°S, Figure 2, response letter). Further, the simulated seasonal pattern and magnitude of the monthly average extreme phytoplankton blooms in the GAB is consistent with remotely sensed observations. Both have a bimodal distribution with an increase in extreme phytoplankton blooms towards September and a maximum in May (Figure 1, response letter). These results help support the study findings regarding phytoplankton blooms.

**The reviewer states that from figure 1 it seems that the model significantly**

**overestimates the drawdown of NO3 in the central part of the study domain. The reviewer recommends exploring the difference since it can dramatically change the relative importance of different NO3 sources. Finally, the reviewer recommends using a sequential colormap.**

Figure 1 in the manuscripts presents modeled and observed vertical profiles (c & d), and depth averaged annual NO3 means (a & b). Annual means of NO3 were depth averaged from the surface to ~4500m. The largest differences between observations and modeled NO3 occur at depth, particularly below 1000m (Figure 1c). Indeed, the discrepancy the reviewer mentions (less NO3 in the modeled GAB-SAFn region) does not occur when the annual mean NO3 is averaged over the first 1000m. We now show 0-1000m average (Figure 4, response letter) because most of the particle trajectories that go into the GAB are distributed between the surface and 1000m. Importantly, this new figure shows the model is generally consistent with the observations. We have changed the colormap and are now using hotter colors for higher values. Figure 4 of this response letter would replace Figure 1 in the manuscript.

[Figure]

Figure 4. a) Depth-averaged (0-1000m) annual mean of NO3 concentration from OFAM-WOMBAT (1996), and b) CARS. c) Annual mean and standard deviation of nitrate concentrations at a location within each source (i.e. Pacific, Southern, Indian Oceans, and GAB-SAFn). Solid lines are model outputs ant dashed lines are CARS observations. In panel (a) the grey line indicates location were particles were seeded along the 2000m isobath; the black lines indicate the boundaries between the different sources (i.e. Pacific, Southern, Indian Oceans, and GAB-SAFn), and the symbols show the distribution of some particles at the nitrate memory time (81days) before entering the GAB exemplifying different sources.

**The reviewer found confusing the explanation of the method used to calculate the decorrelation time of NO3, asks if we consider absolute NO3 concentrations and recommends making this section more explicit.**

Computing the auto-correlation with absolute values would only shift the magnitude of the auto-correlation and the threshold criteria to define the de-correlation time. Removing the mean allows using zero crossing to define the memory time.

Following the reviewer suggestions we explicitly describe (in bold below) the method used to obtain the decorrelation time series within the original text of the manuscript (normal font below) (this would be added in the manuscript).

"To deconstruct the contribution of nitrate from different oceans into the GAB, we quantify the time over which the **auto-correlation** of nitrate concentration of water parcels that reach the GAB **remains positive** along their backward trajectories (i.e. de-correlation time). This can be considered a memory time of nitrate concentration. If the nitrate concentration of a water parcel becomes significantly different over time along its trajectory, it means that the resource has been used or replenished; thus, short memory times imply local resource recycling **while large memory times imply that nitrate has been conserved and is sourced from afar**. The nitrate source of water masses entering the GAB is the location of the water parcel just before the auto-correlation of the nitrate concentration becomes zero. To compute the nitrate memory time or de-correlation time **we first obtain nitrate concentration anomalies for each trajectory: $n_{Tr}'(t) = n_{Tr}(t) - \overline{n}_{Tr}$, where $n_{Tr}$ is nitrate concentration as a function of time $t$ along the backward water parcel trajectory $Tr$. Correlation coefficients ($C$) between nitrate concentration anomalies across all trajectories ($NTr$) are computed as: $C(lt) = \frac{1}{NTr}\sum_{Tr=1}^{NTr}[n'_{Tr}(t)][n'_{Tr}(t - dt * lt)]$ for every time step ($dt$) backwards in time over the duration ($lt$) of the water parcel trajectories before they reach the GAB. This yields auto-correlation coefficients as a function of time where time one corresponds to the time the water parcel enters the GAB;** the last time when the correlation coefficient is larger than zero corresponds to the de-correlation time scale or nitrate memory time, and the water parcel location at that time indicates the source of nitrate".

**4) It would be very useful to compare these values with growth rates from the model, which should be explicitly calculated by the NPZD module.**

Considering the simulated particle trajectories are a good reflection of water movement growth rates explicitly computed with the NPZD module and those obtained by changes in phytoplankton concentrations along trajectories should be similar.

**5) How do you take vertical diffusion of nutrients into account? I would assume that a significant amount of tracer below 100 meters could be transported in ways not picked up by the particle tracking?**

Although vertical diffusion is not implemented in the particle tracking simulations OFAM parameterizes it. The observed and modeled vertical profiles of [NO3] compare well down to 1000m (Figure 1c,d), implying reasonable representation of nutrient vertical distribution and hence vertical diffusion processes. In addition, over the time scales we are interested in (months), vertical diffusion is very small. Using a high vertical diffusion Kz of 10^-4 m2/s, a time scale of 82 days, and the assumption of Brownian motion, the spread due to

diffusion will be $\sqrt{Kz\,dt}$ = 26 m. I.e. over this time scale, diffusion will be only a couple tens of meters.

Moreover larger changes in nutrient concentrations are more likely a consequence of biological activity than vertical diffusion. For instance, Qin *et al.* 2015 show that in the Pacific Ocean, changes in iron concentrations due to vertical diffusion are an order of magnitude smaller than those caused by scavenging. Thus, neglecting further effects of vertical diffusion on nitrate concentrations distributions is reasonable. Similarly, in the oligotrophic water of the region if a parcel of water gets into the mixed layer depth (MLD) it will be almost completely used by the biology and reduced to nearly zero. Thus, its vertical diffusion within the MLD (where is expected to be largest in magnitude) would not have the time to affect nutrient distributions before they are used.

**6) How much of the NO3 in GAB/SAFn as SAF originates from the southern or Indian ocean? It's possible that some of the available nutrients in nearby areas in fact has been transported there from afar.**

Backward trajectories over longer periods of time than the nitrate memory time show that water parcels in the GAB-SAFn region can come from the Southern, Indian, and Pacific oceans. However, given the calculated nitrate memory time (81 days) little nitrate in the GAB-SAFn is sourced from water from outside GAB-SAFn because transporting water from the Indian, Pacific and Southern Ocean often takes longer than 81 days (time by which nitrate would have been used).

*Reviewer 2.*

**The motivation for the study is not clear: either the focus on the most rapid increases in chlorophyll or the consequences of knowing the sources of N consumed in these blooms. Their conclusions are that much of the N for these extreme blooms originates in the upper 100 m of the ocean between the Great Australian Bight and the Sub-Antarctic front. While this is an interesting analysis, the significance is not clearly communicated.**

In the study, we show how one can utilize particle tracking to provide insight into biophysical processes. First, we show the particle tracking can determine the source of nitrate to the GAB for phytoplankton growth. Second, we analyze the track particles to elucidate the temporal and spatial features of extreme phytoplankton blooms and their nitrate sources (we would add this in the discussion).

We would modify the manuscript text to emphasize that the study objective is to examine both the rapid increase in chlorophyll, and the consequence of knowing the nitrate sources at the same time; i.e. to identify temporal (up to inter-annual) and spatial (within the whole GAB) variability of extreme phytoplankton blooms, the sources of nitrate that fuel them, and the implications (i.e. possible mechanisms inducing the supply of nitrate that allows phytoplankton blooms and their long term fluctuations.

In Page 1 line 20. We mention that phytoplankton blooms in the east side of the GAB are known to be induced by upwelling in summer but that longer-term fluctuations as well as the occurrence of blooms elsewhere in the GAB have received little attention. We would extend this statement to explain that the identified summer eastern GAB blooms are induced by upwelling **and the consequent supply of nutrients**. We also emphasize that **not only a thorough characterization of phytoplankton blooms is lacking but also knowledge regarding their nutrient sources**.

We would expect this helping the reader to appreciate the fact that the study examines both phytoplankton blooms and the nutrients sustaining them.

In page 2, line 3 we would include a paragraph to motivate the focus on episodes of extreme increase in phytoplankton (see response to comment below) and would mention that phytoplankton growth is linked to the uptake of nutrients.

In page 2, line 17 we would rephrase the paragraph to emphasize that we study phytoplankton blooms and the sources of nitrate used for the blooms rather than nitrate supply in general.

**key ideas are frequently undeveloped (why focus on extreme blooms)**

We would re-write and extend a previous paragraph (page 2, lines 11-13) justifying the focus on extreme increase in phytoplankton concentrations, In page 2 after line 3 (see the text below in bold).

**"The efficiency of energy transfer up the food change depends on the size distribution of the phytoplankton community (van Ruth), which is in turn related to their rates of growth and nutrient uptake. The growth of large algal cells which allow the transfer of energy to higher trophic levels in fewer steps than small phytoplankton (van Ruth), is likely to induce large increase and decrease of phytoplankton and nutrients concentrations respectively. This type of efficient food chain (i.e. sustained by large phytoplankton) is known to support fisheries globally (Ryther 1969, Teira et al 2001). Thus, the focus of this study is on blooms characterized as episodes of extreme phytoplankton increase and simultaneous nutrient decrease.**

**Although phytoplankton blooms are natural events that are the base of the marine food chain and provide the energy to sustain marine life; harmful algal blooms (HABs), which produce toxic substances, cause anoxic conditions blocking light to organisms lower in the water column, and even clog the gills of fish are also known to occur in the GAB and are often extreme (Hallegraeff et al. 2010, Hallegraeff et al. 2012). Focusing on extreme blooms is therefore revealing plausible characteristics of HABs in the region.**

**Would the conclusions of the paper differ if the authors focused on less extreme phytoplankton blooms? How much of total productivity is in these extreme blooms? (I see this is at page 7, line 10; an earlier motivational mention would be helpful). Blooms take days to weeks to develop, why do you focus on such short periods (large daily changes)? Perhaps large growth rates would be worth a closer look?**

We do not examine non-extreme phytoplankton blooms but we would extend the motivation for focusing on these particular episodes in the introduction (in page 2 after line 3, see comment above). We do not mention how much of the total productivity these episodes represent before showing results because this is an output of the study analysis, but we do mention the proportion relative to the total daily changes in phytoplankton that these episodes represent in the methods section (page 4, lines 22-23), and would provide an extended early motivation for examining extreme events in the introduction (in page 2 after line 3, see response to comment above).

Following the reviewer suggestions we examine extreme changes in phytoplankton concentrations occurring over a one-week period (hereafter weekly phytoplankton blooms). In order to obtain weekly changes of phytoplankton concentrations we low pass filtered (using a Lanczos filter) the time series of phytoplankton and nitrate concentrations that correspond to those along particle trajectories, and resampled them to obtain phytoplankton and nitrate concentrations every 7 days.

The analyses conducted for extreme daily phytoplankton blooms (presented in the manuscript) were then applied to extreme weekly phytoplankton blooms. Specifically, we constructed 12 year long time series of weekly changes in phytoplankton concentrations distinguishing the longitude where the blooms occurred, the nitrate source, and the depth at which the nitrate used in the bloom entered the GAB (Figure S2 in the manuscript for daily phytoplankton blooms, Figure 3 in this response letter for weekly phytoplankton blooms). Time series for both daily and weekly phytoplankton blooms show a clear (dominant) annual periodicity.

[Figure]

Figure 5. Time series (June1993-June2005) of weekly increase in phytoplankton associated with extreme phytoplankton blooms; binned every 10 days and coloured by: a) longitude where the episodes occur, b) nitrate source, and c) depth at which the water entered the GAB.

The phytoplankton bloom climatologies are similar; daily/weekly phytoplankton blooms reveal a bimodal distribution peaking in autumn and spring (May/June and October/September), and the relative contribution to phytoplankton blooms of sources and depths at which the nitrate is supplied into the GAB are similar (mostly from the GAB-SAFn and 55%/60% from water entering the GAB above 100m, Figure 3 in the manuscript for daily phytoplankton blooms and Figure 6 of this review for weekly phytoplankton blooms) .

[Figure]

Figure 6. a) Monthly means of weekly increase in phytoplankton associated with extreme phytoplankton blooms occurring between June 1993 and June 2005; binned by: b) longitude inside the GAB where the extreme phytoplankton blooms occur, c) nitrate source, and d) depth at which the water parcel enters the GAB. Error-bars in panel (a) indicate the standard deviation.

We would include paragraphs in the manuscript (in page 4 after line 28) explaining that weekly phytoplankton blooms were briefly analyzed for comparisons and they show similar temporal fluctuations (annual periodicity and a climatology that reveals a bimodal distribution peaking in autumn and spring), as well as similar relative contribution of nitrate sources (mostly from the GAB-SAFn, followed by the Southern, Indian and Pacific) and relative contribution of water entering above and below the GAB to phytoplankton blooms. These results would be mentioned in page 5 as the first paragraph of the results section 3.2 "Episodes of extreme phytoplankton blooms from simultaneous extreme nitrate decrease and phytoplankton increase".

**I'm going to echo two comments by the other referee: (1) the study is based entirely on model calculations; some connection to remote sensing data or other observations would be most welcome, (2) the description of decorrelation / memory time should be expanded.**
Please see our responses to reviewer 1 comments 1 and 3 where we address these comments.

**Abstract - "novel Lagrangian approach" - what is the value of stressing the novelty? You don't clearly explain why its novel.**

We removed 'novel' (Page 1, line 2)

**"enters the GAB in the upper 100m" – sounds like this is a contrast with "stratification erosion is key supplying nutrients". Stratification erosion does not actually supply nutrients. Are you referring to entrainment of nutrients from below the nutricline? Which is it? Lateral advection or entrainment?**

Yes, we are referring to entrainment. We would rephrase the abstract to clarify that stratification erosion allows the entrainment of nutrients from below the nutricline (in page 1 line 8). We would also modified the text explaining that phytoplankton blooms are sourced by water entering the GAB in the top 100m (55% of the times and most often in spring) but also by water that enters the GAB at depth (mostly in autumn) (modifications in page 1 lines 4-8, in bold below).

"Our results show extreme phytoplankton blooms have annual **and semiannual periodicities**, peaking in the Austral **spring when they are mostly sourced by water entering the GAB above 100m, and in** autumn when **they are mostly sourced by water entering the GAB below 100m and the time of year when** the mixed layer deepens. **Our findings** suggest that stratification erosion is key **process supplying** nutrients **below the nutricline** into the euphotic zone and triggering **phytoplankton blooms**".

**p1, l10. I think "regulates" is an overstatement**.

We would change 'regulates' for 'contributes to the regulation of'

**p1, l15. This is an unusual use of the word "catchment".**

We would use catch.

**p1, l17. "Coastal regions account for a significant.." this is vague. Can you be more precise?**

We would provide the percentages (14-30%) of coastal productivity relative to total ocean productivity reported in the cited references (Pomeroy 1974 and Gattuso 1998)

**p2, l12. "important implications": this is vague. What are the implications of "extreme blooms" (here called episodic) that justify the focus in this study?**

We agree with the reviewer that the statement was vague, we would re-write this paragraph (in page 2 after line 3) and explain that extreme blooms have implications on the efficiency to transfer energy through the food chain. As well as such events may be associated with harmful algal blooms with important ecological and economic consequences for the GAB (see response to previous comment and).

**p2, l18. "first time": the focus on priority rather than your message is not helpful or needed.**

We would remove first time.

**p2, l20. I didn't find a disentangling of the mechanisms later in the paper**

When examining episodes of extreme daily increase in phytoplankton we identified: a) spring blooms in the east and west sides of the GAB often sourced by waters coming from the Indian and Southern Oceans that enter the GAB in the top 100m; and b) autumn blooms in the middle of the GAB mostly sourced by waters from the GAB-SAFn that enter the GAB at depth. This is explained in the results page 6, lines 17-22, and we would include a Figure 5 in the manuscript, figure 7 in this response letter to emphasize them). The fact that phytoplankton blooms sourced by waters from the Indian and Southern Oceans peak in spring is likely a consequence of the Leeuwin current intensification over winter (explained in page 8, lines 7-10) and stronger winds between Antarctica and Australia that induce equatorward Ekman transport from the Southern Ocean in spring (explained in page 8 lines 10-12). Further in the spring light levels are low enough to prevent complete nitrate depletion of the source water and allow such water to be a supply of nitrate to the GAB.

The autumn blooms are mostly associated with water that enters the GAB at depth suggests that the deepening of the mixed layer depth (starting in autumn, particularly in the middle of the GAB), allows the entrainment of deep-water nutrients into the euphotic zone (explained in page 7 lines 21-26 lines). These results show possible processes (i.e. Leeuwin current intensification, wind strengthening between Antarctica and Australia, deepening of the mixed layer) supplying nitrate from specific oceans and provoking phytoplankton blooms in the region. These findings are explained throughout the text and we would include a new Figure 5 in the manuscript to highlight how the link between temporal and spatial variability of phytoplankton blooms and their nitrate sources reveals possible mechanisms driving the temporal and spatial variability of phytoplankton blooms.

[Figure]

Figure 7. Spatial distribution of extreme phytoplankton blooms occurring in a)

spring and b) autumn. Square and circular shapes represent whether the water parcel entered the GAB above or below 100 respectively and colors indicate the source location.

**p3, l10. Year missing from citation.**

We would included the year (2015).

**p3, l25. There is some good detail in this paragraph, but I found that I still have considerable uncertainty about what you did. For example, how is memory time defined and calculated?**

We would try to clarify how is memory time computed in the section by adding a new paragraph to the manuscript starting with ''Nitrate memory time and sources'' (see response to reviewer 1 comment 3 above).

**p3, l28. I didn't understand "prevails", and thus the whole sentence escaped interpretation.**

We would replace "prevails" with "remains positively correlated" . We would explain how such correlation is computed (see response to reviewer 1 comment 3 above).

**p3, l29. This is another vague explanation of "memory time". A rewriting of this paragraph would be helpful.**

We would re-write this paragraph (see response to above comments).

**p4, l9. Do you really mean that the concentration gradient has a peak (trough?) here? Why does light create a sudden increase (decrease?) in the concentration gradient?**

We would change "peak" for "trough" to correctly indicate that the gradient is negative (decrease in phytoplankton concentration with depth). We would rephrase the next sentence to explain that this is caused by a decrease in light below the minimum necessary for phytoplankton growth (Page 4 line 10).

**p4, l11. "were" should be "where"**

We would change "were" for "where"

**p4, l14. What do you mean by "efficiency" here?**

We would explain that by efficiency we refer to the capacity to transform nitrate in phytoplankton biomass at a given temperature and light intensity.

**p5, l25. I don't think you need "respectively" here**.

The reviewer is correct; we would remove 'respectively'.

**p6, l10-15. I found this difficult to follow. if 90% of the N is disappearing, does that imply rapid grazing tightly coupled to very short term increases (daily) in phytoplankton biomass? Are you sure your grazing model is good enough to draw these conclusions and make them interesting?**

We would modify this paragraph to clarify that discrepancy between the magnitudes of decrease in nitrate concentrations and increase in phytoplankton concentrations are a consequence of low efficiency of phytoplankton to convert nitrate into biomass without referring to grazing processes. Nevertheless, the GAB is an oligotrophic region where phytoplankton growth and grazing tend to be closely coupled. Generally under such conditions the phytoplankton growth is rapidly grazed preventing nitrate draw-down to be associated with a large phytoplankton bloom. The occurrence of such conditions are closely linked to the initial phytoplankton and zooplankton concentrations and how ideal the conditions are for phytoplankton growth. To get a phytoplankton bloom, phytoplankton growth must exceed zooplankton grazing.

**p8, l2. You compare to the effects of winds here, but I didn't see where you analyzed winds in your methods + results.**

Correct, this study does not examine wind stress directly. We found phytoplankton blooms in summer; however, they are smaller in magnitude and less frequent than those in spring and autumn (source water is different too). Given the difference in magnitude and frequency between extreme phytoplankton blooms in summer from the spring and autumn blooms we postulate different processes cause them. Summer phytoplankton blooms in the eastern side of the GAB are caused by wind driven upwelling events (e.g. Kampf et al. 2004) and therefore it is consistent to have extreme summer blooms associated with these episodic wind-driven upwelling events. While in the spring and autumn, the mixed layer depth is most variable and the entrainment of nutrients into the euphotic zone is a more likely mechanism to drive extreme phytoplankton blooms in the GAB.

**p8, l13-16. I found the focus on interannual features, which were not discussed elsewhere in the paper, to be jarring and confusing at this point. Might be very interesting / important, but should appear earlier, with some evidence or mention of analysis**

We agree with the reviewer that there was no previous mention of analysis of long-term variability of significant periodicities of phytoplankton blooms. We would include a paragraph in the introduction (in page 2 after line 22), in the methods (section 2.5 'Temporal variability" in page 5 line 6) and results (section "Temporal variability" page 6 line 29) sections to make the reader aware of the motivation, execution (emphasizing the temporal analysis in 12 year-long time series), and outcomes of such analysis before discussing them.
New text included below in bold amongst the old text. In the introduction:
**"In addition, currents that feed water into the GAB and their nutrient concentrations fluctuate over different timescales. For instance, El Nino**

**Southern Oscillation (ENSO) modulates the strength of the typically low-nutrient Leeuwin current (Feng *et al.* 2003) that flows into the GAB through its western and south boundaries (Middleton *et al.* 2014). Similarly, Ayers and Strutton (2013) found higher nutrient concentrations in waters of the Southern Ocean that eventually reach the GAB during the positive phase of the decadal Southern Annular Mode (SAM). However, there is little understanding of the influence of seasonal and longer-term fluctuations of currents that feed water into the GAB on the supply of nutrients used by phytoplankton to grow in this region."** ...

..."By relating the extreme phytoplankton blooms to the sources supplying nitrate, and characterizing their spatial and temporal variability over a 12-year period, we can disentangle which mechanisms may induce extreme blooms inside the GAB **and elucidate the impact of circulation long-term fluctuations or future climate induced changes on extreme phytoplankton blooms**."

In the methods"

"Wavelet analysis on each constituent of the time series is conducted to reveal periodicities linked to specific regions inside the GAB where the extreme phytoplankton blooms occurred, nitrate sources, and depths at which the water parcels enter the GAB (Figure 4, Figure S4) **and how these change over long time scales (12 years)**".

In the results:

**"Temporal wavelets of extreme phytoplankton blooms time series reveal changes in the significance of annual periodicities over time that depend upon the nitrate source. Particularly, blooms associated with nitrate sourced from the Southern Ocean have a significant annual periodicity from 2000 onwards, while those associated with nitrate sourced from the Indian Ocean have a significant annual periodicity only between 1998 and 2001."**